# DC-LLM: Hardware-Friendly LLM Weight Compression via Dynamic Linear Combination

## Abstract

The progressive scaling of large language models (LLMs) has consistently enhanced multimodal understanding and advanced reasoning capabilities, but has substantially increased computational and hardware execution overhead. In this paper, we present DC-LLM, a novel post-method that compresses only model weights. We partition each weight tensor into fixed-size blocks and assign a single seed to each block. The seed drives a hardware-friendly Linear Feedback Shift Register (LFSR) generator that dynamically produces multiple basis matrices. Each block is then reconstructed as a linear combination of these basis matrices, with block-specific coefficients, which substantially reduces the amount of stored data, increases the data-transfer efficiency between memory and compute units, and consequently speeds up memory-bound inference for large language models. Experimental results on different LLM models ranging from 7B–70B parameters show that DC-LLM attains state-of-the-art performance when weights are compressed to approximately 3-bit or 4-bit. We also design a dedicated ASIC accelerator that achieves a 4× speed-up for memory-bound LLM inference.

## 1 Introduction

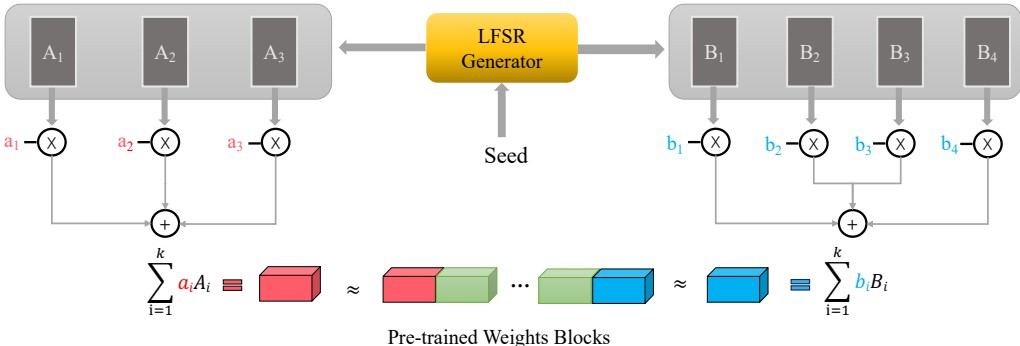

Figure 1: DC-LLM Framework.

Large language models (LLMs) deliver state-of-the-art results across a wide range of natural language processing tasks, and their strong language understanding has been extended successfully to multimodal problems Touvron et al. (2023); Zhang et al. (2022). However, their large computational and memory footprints remain a major barrier to practical use. For example, GPT-3 Brown et al. (2020), with roughly 175 billion parameters, requires on the order of 350 GB of memory when stored in FP16, which effectively translates into the need for at least five NVIDIA A100 80GB GPUs to perform inference. The resulting computation and inter-GPU communication overheads make real-world deployment costly and technically challenging.

Autoregressive large language model (LLM) inference is primarily constrained by memory bandwidth, as the retrieval of weight and activations dominates execution time. Off-chip DRAM accesses entail orders-of-magnitude greater latency and energy consumption compared to on-chip multi-

ply–accumulate (MAC) operations. Consequently, reducing the memory footprint through model compression represents the most effective strategy for mitigating inference latency, lowering power consumption, and reducing deployment costs. Quantization Xiao et al. (2023) represents model weights and activations using lower-precision formats to decrease storage and bandwidth demands. Pruning Ma et al. (2023); Sun et al. (2023) removes parameters judged to be redundant, reducing model size and computational cost. However, most post-training compression methods rely on calibration data and suffer severe accuracy degradation under extreme compression. So, we explore one question whether a calibration-free compression method can be designed that maintains acceptable accuracy at extreme compression?

We present DC-LLM, a weight-only compression method that achieves extreme compression—approximately 3-bit effective precision—while maintaining acceptable accuracy. DC-LLM partitions each weight matrix into fixed-size blocks and approximates each block with a small set of basis tensors that we deterministically generate from a seed. We reconstruct a block by linearly combining the generated basis tensors and multiplying each basis by an optimal coefficient. As a result, we represent and transmit a block's parameters by a single seed and its coefficient vector, which substantially reduces storage and communication bandwidth compared with storing raw weights. In contrast to a previous work SeedLM Shafipour et al. (2024), which reconstructs each block using floating-point matrix multiplications and incurs significant hardware overhead, DC-LLM avoids expensive matrix operations. As a result, DC-LLM achieves higher accuracy while substantially reducing hardware cost.

DC-LLM has two practical challenges. First, weight blocks exhibit large numerical variability, so we must determine how many basis tensors to generate for each block. Second, design parameters such as block size and seed length trade off against post-compression accuracy and average bits per weight, so we must find a Pareto balance among these objectives. To address the first challenge, we define the explained energy ratio to measure the fraction of a block's energy retained by a given basis set, and we use this metric to adaptively select the number of basis tensors per block. To tackle the second challenge, we formulate the selection of block size, seed length, and associated hyperparameters as a multi-objective design-space exploration problem. We then employ Bayesian optimization to identify operating points that achieve a desirable trade-off among accuracy, compression rate.

DC-LLM increases on-chip computation within bounded limits to reduce off-chip memory accesses and to improve effective chip-to-chip bandwidth, enabling extreme compression in multi-chip deployments. The tensor generator uses a linear-feedback shift register (LFSR), a communications-domain primitive Win & Kyaw (2008) that relies primarily on hardware-friendly shift and XOR operations. This choice yields a compact, deterministic, and easily pipelined hardware implementation.

**We make the following contributions in this paper:**

- We propose a novel weight-only compression method, DC-LLM, which dynamically reconstructs each weight block from a seed using a Linear-Feedback Shift Register (LFSR) generator, substantially reducing stored information and increasing effective memory bandwidth.

- We adapt the number of basis tensors per block based on each block's explained energy and reconstruction error to balance compression and accuracy.

- We introduce an offline search strategy that employs Bayesian optimization to find optimal configuration parameters such as block size and seed length.

- We designed a custom hardware accelerator, implemented in SystemVerilog, and demonstrated in simulation that for memory-bound LLM inference it achieves up to a 4× speedup.

- Extensive experiments on LLaMA 2 and LLaMA 3 models ranging from 7B–70B parameters show that DC-LLM attains state-of-the-art performance with weights compressed to approximately 3-bit or 4-bit.

## 2    RELATED WORK

### 2.1    LLM WEIGHT COMPRESSION METHODS

Weight-only quantization targets representing model weights at reduced bit widths to lower storage and compute requirements. For example, GPTQ Frantar et al. (2022) uses block-wise reconstruction

to attain 3–4 bit quantization. SpQR Dettmers et al. (2023b), OWQ Lee et al. (2024), and AWQ Lin et al. (2024) prioritize weights associated with large-magnitude activations. Consequently, SpQR and OWQ adopt mixed-precision schemes to preserve those critical weights, while AWQ applies channel-wise scaling to avoid the hardware inefficiencies of mixed precision. Qlora Dettmers et al. (2023a) recovers performance by performing parameter-efficient fine-tuning on the quantized model. In QuIP# Tseng et al. (2024), Hessian analysis of calibration data helps make rounding decisions during quantization.

LLM pruning has emerged as a critical challenge as large language models continue to scale in size. Conventional pruning techniques, which typically involve retraining the entire model, are computationally expensive and increasingly infeasible for models of this magnitude. Recent work has shifted toward post-training pruning approaches Frantar & Alistarh (2023); Sun et al. (2023); Das et al. (2023), where specialized scoring functions are employed to assess the significance of weights and prune less influential components without requiring costly retraining. In addition, SliceGPT Ashkboos et al. (2024) advances structured pruning by eliminating rows or columns of weight matrices according to eigenvectors and eigenvalues derived from the input, thereby offering a more principled strategy for reducing model complexity.

## 2.2 COMPRESSION WITH PSEUDO-RANDOM GENERATOR

Recent work shows that network weights can be compactly represented by a pseudo-random generator seed together with compact coefficient vectors. PRANC Nooralinejad et al. (2023) compresses entire networks by orders of magnitude to reduce storage and improve transmission efficiency. LoRA Hu et al. (2022) lowers weight storage by injecting trainable low-rank decomposition matrices into each layer. NOLA Koohpayegani et al. (2023) builds on LoRA by expressing low-rank factors as linear combinations of random basis vectors, further reducing memory footprint and computational overhead. SeedLM Shafipour et al. (2024) is first use pseudo-random generator in LLM weight compression, but block reconstruction relies on floating-point multiplications between basis tensors and their coefficients, significantly increasing power consumption and silicon area.

## 3 METHODOLOGY

### 3.1 WEIGHT COMPRESSION USING LINEAR FEEDBACK SHIFT REGISTER GENERATOR

A Linear Feedback Shift Register (LFSR) is a compact and efficient type of shift register that is widely used to produce pseudo-random binary sequences. Its hardware implementation is highly attractive due to its low cost, minimal power consumption, and reliance solely on shift registers combined with XOR logic. These properties make LFSRs suitable for scenarios that require efficient pseudo-random sequence generation, such as signal processing and data compression.

The behavior of an LFSR is determined by two key elements: the register length $K$ and its associated feedback polynomial. During each update cycle, all bits in the register are shifted one position to the right, and the most significant bit is replaced with a new bit computed from a linear combination of selected register bits. This new bit is derived according to the feedback polynomial through modulo-2 arithmetic, which corresponds to XOR operations. Mathematically, the next bit can be expressed as

$$x_{n+1} = \sum_{i=0}^{K-1} \alpha_i \cdot x_{n+i-K+1} \pmod 2, \qquad (1)$$

where $\alpha_i \in \{0, 1\}$ represents the feedback coefficients that determine which register bits participate in the XOR computation.

Since the register contains only a finite number of states ($2^K$ in total), the sequence generated by an LFSR will inevitably enter a repeating cycle. A special case is a maximal-length LFSR, which can traverse $2^K - 1$ nonzero states before repeating. This property is achieved when the feedback polynomial is primitive over the Galois field $GF(2)$, which guarantees that every nonzero state is visited exactly once before the sequence cycles.

In practical applications, precomputing all possible LFSR states for a fixed $K$ and its feedback coefficients $\{\alpha_j\}$ can significantly improve efficiency. By storing the full sequence of $2^K - 1$ states,

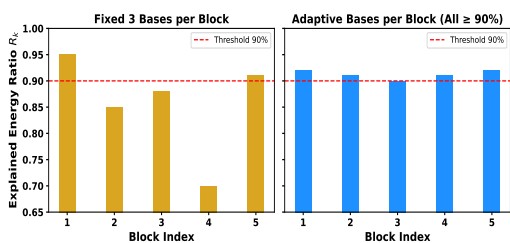
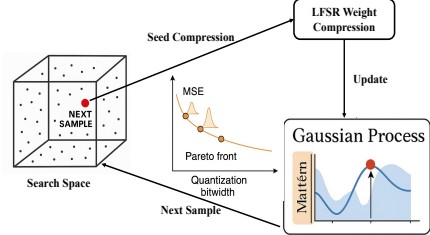

Figure 2: Fixed vs. Adaptive Bases.    Figure 3: Design Space Exploration.

one can generate pseudo-random numbers or populate random matrices without recalculating the shift register at every step. This caching approach provides a highly scalable and memory-efficient mechanism for large-scale pseudo-random sequence generation, as the storage requirement grows linearly with the number of states and remains negligible for moderate register lengths.

We represent each model weight block as a linear combination of normalized sub-blocks generated from a pseudo-random seed using an LFSR-based sequence generator. Let $V(s)$ denote the raw integer sequence of length $L$ generated from a seed $s$. To facilitate stable weight reconstruction, we first center and normalize this sequence into $[-1, 1]$ as

$$U(s) = \frac{1}{2^{K-1} - 1}\Big(V(s) - 2^{K-1}\mathbf{1}\Big), \tag{2}$$

where $\mathbf{1}$ is an all-one vector of the same shape as $V(s)$, and $K$ is the register length of the LFSR. This step ensures that each generated block has zero-centered and normalized values.

The original weight tensor $W$ is partitioned into $q$ blocks. Given a normalized sequence, the $n$-th weight block $\hat{w}_n$ is synthesized by linearly combining $M$ consecutive normalized sub-blocks:

$$\hat{w}_n = \sum_{i=1}^{M} a_{n,i}\, U_i(s_n), \qquad n = 1, 2, \ldots, q, \tag{3}$$

where $a_{n,i}$ are the corresponding scaling coefficients, and $U_i(s_n)$ denotes the $i$-th normalized sub-block generated from seed $s_n$.

The complete weight tensor is reconstructed by stacking all synthesized blocks:

$$\hat{W} = \big[\, \hat{w}_1,\ \hat{w}_2,\ \ldots,\ \hat{w}_q \,\big], \tag{4}$$

where $\hat{W}$ serves as a compact approximation of the original model weights, parameterized by the seeds and their associated coefficients.

## 3.2 Explained Energy and Reconstruction Error

When a weight tensor is approximated using a limited set of basis tensors, a reconstruction error inevitably arises due to the projection onto a low-dimensional subspace. Let $T$ denote the original weight tensor (or a flattened block), and let $\hat{T}^{(k)}$ denote its approximation using $k$ selected bases. The reconstruction error is naturally measured by the squared Frobenius norm

$$\mathcal{E}_k = \big\| T - \hat{T}^{(k)} \big\|_F^2, \tag{5}$$

which captures the energy of the residual orthogonal to the selected subspace.

To quantitatively assess how much of the original weight energy is preserved after compression, we define the explained energy ratio as

$$R_k = \frac{\|P_k T\|_F^2}{\|T\|_F^2} = 1 - \frac{\|T - \hat{T}^{(k)}\|_F^2}{\|T\|_F^2}, \tag{6}$$

where $P_k$ denotes the orthogonal projection operator onto the $k$-dimensional subspace spanned by the selected bases. A higher $R_k$ indicates that most of the tensor energy is captured, corresponding to a lower reconstruction error.

**Algorithm 1:** Adaptive Basis Selection and Quantization

---

**Input:** Weight blocks $\{w_i\}$, energy threshold $R_{\text{th}}$, blocks per quantization $G$
**Output:** Seeds $\{s^*\}$, basis counts $\{k^*\}$, quantized coefficients $\{\hat{a}\}$
**foreach** *block $w_i$* **do**
    $k \leftarrow 2$;
    **repeat**
        **foreach** *seed $s$* **do**
            $a_i \leftarrow U_k(s)^{\dagger} w_i$;
            Compute $R_k(s)$ using Eq. 6;
        $s^* \leftarrow \arg\max_s R_k(s)$;
        **if** $\max_s R_k(s) < R_{\text{th}}$ **then**
            $k \leftarrow k + 1$;
    **until** $\max_s R_k(s) \geq R_{\text{th}}$;
    Record $s^*$, $k$, and $a_i$;
Group every $G$ blocks and quantize $\{a_i\}$ to int8 with a shared scale;
**return** $\{s^*\}$, $\{k^*\}$, $\{\hat{a}\}$;

---

This formulation is grounded in the orthogonal projection theorem in linear algebra, which guarantees that the best low-dimensional approximation is achieved via orthogonal projection and that the residual energy is orthogonal to the selected subspace Strang (2016); Trefethen & Bau III (1997). Moreover, $R_k$ is conceptually analogous to the explained variance ratio commonly used in Principal Component Analysis (PCA) to evaluate how much of the original data variance is preserved in a low-dimensional embedding Jolliffe (2002); Golub & Van Loan (2013).

### 3.3 ADAPTIVE BASIS SELECTION AND QUANTIZATION

As shown in Fig. 2 (left), using a fixed number of bases for all weight blocks leads to inefficient and uneven reconstruction quality. Some blocks are over-allocated, achieving energy ratios far above the target threshold $R_{\text{th}}$ and wasting bases, while other blocks fail to reach the threshold, resulting in poor reconstruction. To overcome this limitation, we adopt an adaptive strategy that dynamically selects the number of bases for each block to reach the target explained energy ratio.

**Algorithm Overview.** For each weight block $w$, we start with a small number of bases ($k = 2$) and enumerate all LFSR seeds $s$. For each candidate seed, we construct the basis matrix $U_k(s) \in \mathbb{R}^{B \times k}$, whose columns are the $k$ normalized sub-blocks generated from seed $s$ according to Eq. 2. We compute the projection coefficients

$$a = U_k(s)^{\dagger} w, \tag{7}$$

where $^{\dagger}$ denotes the Moore–Penrose pseudo-inverse, and evaluate the explained energy ratio using Eq. 6. If the best $R_k(s)$ is below the threshold $R_{\text{th}}$, we increment $k$ and repeat the search process. Once the threshold is reached, we record the selected seed, basis count, and coefficients. Finally, coefficients of multiple blocks are grouped and quantized to int8 with a shared scaling factor. The detailed procedure is summarized in **Algorithm 1**.

Using this algorithm, as illustrated in Fig. 2 (right), blocks that are easier to approximate automatically use fewer bases, while harder blocks receive more bases. This ensures that all blocks reach the target $R_{\text{th}}$ without redundant basis allocation.

### 3.4 MULTI-OBJECTIVE DESIGN SPACE EXPLORATION

Algorithm 1 compresses the original weights $W$ into $\hat{W}$ under a given configuration by performing adaptive basis selection and coefficient quantization. Each iteration of the algorithm inherently yields two optimization objectives:

1. **Reconstruction loss (MSE)**

$$\mathcal{L}_{\mathrm{MSE}} = \frac{1}{N} \sum_{i=1}^{N} \big\| W_i - \hat{W}_i \big\|_F^2, \qquad (8)$$

where $W_i$ and $\hat{W}_i$ denote the original and reconstructed weight blocks.

2. **Quantization bitwidth**

$$Bitwidth = S + 8k + \frac{16}{G}, \qquad (9)$$

where $S$ is the LFSR seed length, $k$ is the number of adaptive bases for this block, and $G$ is the number of blocks sharing one FP16 scaling factor.

These two metrics inherently form a trade-off: reducing the bitwidth typically increases the MSE loss, while allocating more bases reduces the error but consumes more storage. Crucially, this trade-off is fully determined by a four-dimensional configuration:

$$T = \langle B,\ S,\ G,\ R_{\mathrm{th}} \rangle, \qquad (10)$$

where

- $B$ (**block size**): number of weights per block;
- $S$ (**seed length**): bit length of the LFSR seed;
- $G$ (**blocks per quantization**): number of blocks sharing one FP16 scaling factor;
- $R_{\mathrm{th}}$ (**energy ratio threshold**): target explained energy ratio for adaptive basis selection.

Therefore, the problem of finding a good compression strategy naturally transforms into a multi-objective design space exploration (DSE) problem: each configuration $T$ produces a pair $\big(\mathcal{L}_{\mathrm{MSE}}(T),\ Bitwidth(T)\big)$, and our goal is to search for configurations that achieve a favorable trade-off between accuracy and storage efficiency. To efficiently explore this discrete-continuous search space, we adopt a Bayesian Optimization (BO) framework(as shown in Fig. 3) with a Gaussian Process (GP) surrogate model using a Matérn kernel. At each iteration, the next configuration $T^*$ is selected by maximizing the Expected Incremental Predictive Volume (EIPV) Shah & Ghahramani (2016):

$$T^* = \arg\max_{T \in \mathcal{D}}\ \mathrm{EIPV}(T \mid \mathcal{D}), \qquad (11)$$

where $\mathcal{D}$ denotes the design space, which contains all possible candidate configurations. A configuration $T \in \mathcal{D}$ corresponds to a specific design point and EIPV function is:

$$\mathrm{EIPV}(T \mid \mathcal{D}) = \mathbb{E}_{\mathbf{f}(T)} \big[ \Delta\mathrm{HV}\big(\mathbf{f}(T), \mathcal{P}\big) \big], \qquad (12)$$

$\Delta\mathrm{HV}$ denotes the hypervolume improvement achieved by augmenting the current Pareto set $\mathcal{P}$ with the candidate solution $\mathbf{f}(T)$.

### 3.5 HARDWARE ANALYSIS

We propose a seed-compression-aware accelerator because conventional GPUs, which are highly optimized for dense matrix-multiplication kernels, cannot efficiently reconstruct weights from compact seeds. We implement the Weight Generator and the Systolic Array in SystemVerilog and synthesize the designs with Synopsys Design Compiler Ultra (2017) targeting a 7 nm FinFET Clark et al.

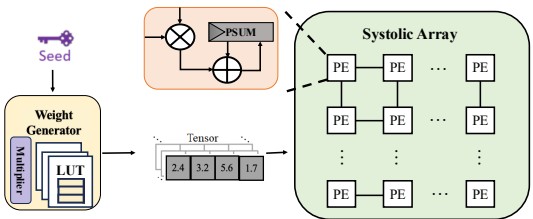

Figure 4: Weight Generator and Systolic Array.

(2016) standard-cell library to obtain

area, timing, and power estimates for the hardware implementations. A cycle-accurate simulator is developed to evaluate end-to-end system performance, and CACTI Muralimanohar et al. (2009) is employed to model on-chip memory latency and power. The Weight Generator comprises a lookup table (LUT) that stores all LFSR states together with the multipliers required to reconstruct weights from seeds. The Systolic Array consists of processing elements (PEs), each integrating a multiplier, an adder, and a partial-sum buffer to accumulate intermediate results.

## 4 EXPERIMENTS

### 4.1 PERFORMANCE ANALYSIS

We evaluate DC-LLM by measuring perplexity on the WikiText-2 benchmark Merity et al. (2016) and running a suite of zero-shot tasks with the LM Evaluation Harness Gao et al. (2021). We compare DC-LLM to SeedLM Shafipour et al. (2024), AWQ Lin et al. (2024), OmniQuant Shao et al. (2023), and QuIP# Tseng et al. (2024), using the official GitHub releases for each project. SeedLM serves as a calibration-free reference, while several other baselines require per-layer calibration on held-out examples. DC-LLM likewise operates without calibration and, as our results show, achieves lower perplexity across multiple model scales and aggressive bitwidths. For AWQ and OmniQuant we apply 4-bit integer quantization with channel-wise scaling to prevent the effective bits per parameter from increasing (a

| Method | Bits | 2-7B | 2-13B | 2-70B | 3-8B | 3-70B |
|---|---|---|---|---|---|---|
| Baseline | 16 | 5.5 | 4.9 | 3.3 | 6.1 | 2.9 |
| DC-LLM (Ours) | 3.8 | 5.7 | 5.0 | 3.5 | 6.8 | 3.6 |
| SeedLM | 4 | 5.7 | 5.1 | 3.5 | 7.0 | 3.8 |
| OmniQuant | 4 | 6.1 | 5.2 | 3.7 | inf | inf |
| AWQ | 4 | 5.8 | 5.1 | 3.5 | 7.1 | 4.7 |
| QuIP# | 4 | 6.5 | 5.3 | OOM | 7.6 | OOM |
| DC-LLM (Ours) | 2.7 | 6.5 | 5.8 | 3.8 | 9.7 | 5.4 |
| SeedLM | 3 | 6.6 | 5.8 | 4.0 | 10.1 | 5.7 |
| OmniQuant | 3 | inf | 10.7 | 7.5 | inf | inf |
| AWQ | 3 | 15.6 | 6.5 | 4.4 | 11.8 | 11.6 |
| QuIP# | 3 | 10.8 | 5.7 | OOM | 10.1 | OOM |

Table 1: We report WikiText-2 perplexities for Llama 2 and Llama 3 using 3- and 4-bit weight representations evaluated on 2048-token contexts. We write model identifiers as `x-yB` to indicate Llama version `x` with `y` billion parameters (for example, `2-7B` denotes Llama 2 with 7B parameters). We record any perplexity value above 100 as `inf` to signal numerical divergence. We emphasize the smallest perplexity per column to call out the best-performing configuration. We mark runs that exceed the memory capacity of four A100 80GB GPUs with `OOM`.

group size of 128 adds roughly 0.25 extra bits per parameter). We avoid fine-tuning the quantized checkpoints for QuIP# and OmniQuant to preserve a fair comparison with the calibration-free workflows used by SeedLM and DC-LLM. To quantify overall language-model fidelity we compute perplexity on the WikiText-2 test. Table 1 reports these measurements and highlights the trade-off between compression aggressiveness and model quality.

We evaluated zero-shot accuracy on a suite of benchmark tasks and summarize the outcomes in Table 2, where DC-LLM matches or surpasses contemporary quantization methods at the same bit budgets. DC-LLM achieves these results without relying on any calibration examples, demonstrating a calibration-free advantage. Collectively, these findings demonstrate that DC-LLM sustains strong multi-task robustness even when applied to models of substantial expressive capacity.

### 4.2 HARDWARE ANALYSIS

In this work we perform hardware-level experiments to compare three weight-handling strategies—weights without compression, 4-bit weight quantization, and DC-LLM—across Llama2 and Llama3 model families ranging from 7B to 70B parameters. We present a systematic comparison of the three methods with respect to latency, energy efficiency, and silicon area.

**Area.** We analyzed the area of the weight generator, which comprises several multipliers and a cache-backed look-up table (LUT) that stores all linear-feedback shift register (LFSR) states. Next, we evaluated the area of the systolic array: each processing element (PE) contains a multiplier, an

| Model | Method | Bits | ARC-Easy | ARC-Challenge | HellaSwag | WinoGrande | BoolQ | Mean |
|---|---|---|---|---|---|---|---|---|
| Llama 2 7B | Baseline | 16 | 74.58 | 46.33 | 75.98 | 69.06 | 77.74 | 68.74 |
| | DC-LLM(Ours) | 3.8 | 73.36 | 44.55 | 74.51 | 68.47 | 77.34 | 67.65 |
| | SeedLM | 4 | 73.23 | 44.54 | 74.45 | 68.43 | 77.19 | 67.57 |
| | AWQ | 4 | 70.58 | 43.94 | 74.96 | 68.75 | 78.29 | 67.30 |
| | QuIP# | 4 | 68.35 | 39.85 | 72.40 | 65.59 | 75.14 | 64.27 |
| | OmniQuant | 4 | 70.71 | 43.52 | 74.20 | 68.27 | 73.64 | 66.07 |
| | DC-LLM(Ours) | 2.7 | 70.01 | 41.39 | 70.74 | 66.38 | 74.29 | 64.56 |
| | SeedLM | 3 | 69.87 | 41.21 | 70.72 | 66.30 | 74.28 | 64.48 |
| | AWQ | 3 | 53.37 | 33.62 | 56.66 | 61.09 | 57.58 | 52.46 |
| | QuIP# | 3 | 59.51 | 34.22 | 59.23 | 61.09 | 65.20 | 55.85 |
| | OmniQuant | 3 | 35.69 | 25.77 | 35.48 | 52.88 | 42.48 | 38.46 |
| Llama 2 13B | Baseline | 16 | 77.44 | 48.98 | 79.38 | 72.22 | 80.55 | 71.71 |
| | DC-LLM(Ours) | 3.8 | 77.02 | 49.93 | 78.55 | 72.81 | 79.33 | 71.53 |
| | SeedLM | 4 | 76.98 | 49.83 | 78.54 | 72.77 | 79.20 | 71.46 |
| | AWQ | 4 | 77.44 | 49.32 | 78.57 | 71.90 | 78.47 | 71.14 |
| | QuIP# | 4 | 74.24 | 45.48 | 77.17 | 71.27 | 79.51 | 69.53 |
| | OmniQuant | 4 | 76.18 | 47.95 | 78.10 | 72.14 | 81.77 | 71.23 |
| | DC-LLM(Ours) | 2.7 | 72.96 | 45.43 | 74.62 | 71.51 | 78.81 | 68.67 |
| | SeedLM | 3 | 72.85 | 45.39 | 74.50 | 71.35 | 78.81 | 68.58 |
| | AWQ | 3 | 70.58 | 45.14 | 72.72 | 64.96 | 72.45 | 65.17 |
| | QuIP# | 3 | 73.48 | 45.14 | 74.92 | 69.06 | 79.60 | 68.44 |
| | OmniQuant | 3 | 55.85 | 34.47 | 59.54 | 53.04 | 63.39 | 53.26 |
| Llama 2 70B | Baseline | 16 | 80.98 | 57.25 | 83.81 | 77.98 | 83.70 | 76.74 |
| | DC-LLM(Ours) | 3.8 | 81.30 | 56.54 | 83.04 | 76.75 | 82.45 | 76.02 |
| | SeedLM | 4 | 81.14 | 56.40 | 82.97 | 76.72 | 82.26 | 75.90 |
| | AWQ | 4 | 80.98 | 56.66 | 83.24 | 77.19 | 83.27 | 76.27 |
| | QuIP# | 4 | OOM | OOM | OOM | OOM | OOM | OOM |
| | OmniQuant | 4 | 79.59 | 55.97 | 82.67 | 76.80 | 83.43 | 75.69 |
| | DC-LLM(Ours) | 2.7 | 79.07 | 53.86 | 80.53 | 76.97 | 79.14 | 73.91 |
| | SeedLM | 3 | 79.00 | 53.84 | 80.51 | 76.80 | 79.02 | 73.83 |
| | AWQ | 3 | 80.26 | 55.80 | 80.50 | 73.01 | 80.00 | 73.91 |
| | QuIP# | 3 | OOM | OOM | OOM | OOM | OOM | OOM |
| | OmniQuant | 3 | 63.59 | 39.51 | 68.24 | 62.04 | 65.23 | 59.72 |
| Llama 3 8B | Baseline | 16 | 76.81 | 52.73 | 76.97 | 72.93 | 81.87 | 72.26 |
| | DC-LLM(Ours) | 3.8 | 76.68 | 49.89 | 76.72 | 73.12 | 80.84 | 71.45 |
| | SeedLM | 4 | 76.52 | 49.74 | 76.61 | 72.93 | 80.76 | 71.31 |
| | AWQ | 4 | 74.49 | 51.54 | 78.03 | 73.09 | 80.40 | 71.51 |
| | QuIP# | 4 | 72.39 | 46.93 | 75.93 | 71.82 | 79.24 | 69.26 |
| | OmniQuant | 4 | 73.95 | 47.78 | 73.42 | 69.69 | 71.99 | 67.37 |
| | DC-LLM(Ours) | 2.7 | 67.32 | 41.72 | 68.46 | 69.39 | 67.73 | 62.92 |
| | SeedLM | 3 | 67.21 | 41.55 | 68.34 | 69.22 | 67.61 | 62.79 |
| | AWQ | 3 | 64.90 | 40.19 | 68.40 | 65.04 | 74.62 | 62.63 |
| | QuIP# | 3 | 65.07 | 40.36 | 67.79 | 68.82 | 72.14 | 62.84 |
| | OmniQuant | 3 | 30.26 | 22.53 | 28.96 | 49.33 | 48.47 | 35.91 |
| Llama 3 70B | Baseline | 16 | 85.23 | 64.33 | 84.07 | 77.66 | 86.27 | 79.51 |
| | DC-LLM(Ours) | 3.8 | 83.94 | 59.31 | 83.89 | 77.80 | 85.62 | 78.11 |
| | SeedLM | 4 | 83.80 | 59.30 | 83.84 | 77.74 | 85.60 | 78.06 |
| | AWQ | 4 | 80.98 | 57.94 | 82.84 | 60.54 | 79.39 | 72.34 |
| | QuIP# | 4 | OOM | OOM | OOM | OOM | OOM | OOM |
| | OmniQuant | 4 | 25.13 | 26.54 | 26.36 | 51.38 | 37.83 | 33.45 |
| | DC-LLM(Ours) | 2.7 | 78.50 | 52.24 | 80.83 | 77.48 | 84.66 | 74.74 |
| | SeedLM | 3 | 78.45 | 52.22 | 80.77 | 77.35 | 84.59 | 74.68 |
| | AWQ | 3 | 65.87 | 45.14 | 70.76 | 55.88 | 69.08 | 61.35 |
| | QuIP# | 3 | OOM | OOM | OOM | OOM | OOM | OOM |
| | OmniQuant | 3 | 25.21 | 25.94 | 26.15 | 49.64 | 37.83 | 32.95 |

Table 2: Performance comparison across different models and zero-shot tasks for around 4-bit and 3-bit configurations. Entries that ran out of memory in our setup are marked with OOM.

adder, and a local buffer. As shown in Table 3, for a $16 \times 16$ systolic array accelerator, the weight generator occupies only a small fraction of the overall area—approximately $3\%$. This indicates that the area overhead introduced by the generator in DC-LLM is negligible.

**Latency and Energy.** During latency and energy measurements, we compared DC-LLM (the average 3.8-bit case) against two baselines: an uncompressed weight design and a 4-bit weight-

| Module | Number | Area (mm$^2$) | Ratio (%) |
|---|---|---|---|
| PE (121.53 $\mu$m$^2$) | 256 | 0.0311 | 97.2% |
| Generator (912.23 $\mu$m$^2$) | 1 | 0.0009 | 2.8% |

Table 3: Area breakdown.

quantization design. To ensure an area-fair comparison among the three configurations, we reduced the number of PEs in DC-LLM so that the combined area of the generator plus the diminished PE array matches the total area of the other two cases. As shown in Figure 5a, relative to the uncompressed baseline DC-LLM achieves an approximately $4\times$ reduction in latency across multiple benchmarks and also outperforms the 4-bit quantized design in execution time. Figure 5b reports energy consumption: weight compression substantially reduces memory-access power, producing large savings in DRAM and SRAM energy; the figure also presents a breakdown of power into static, DRAM, SRAM, and core components.

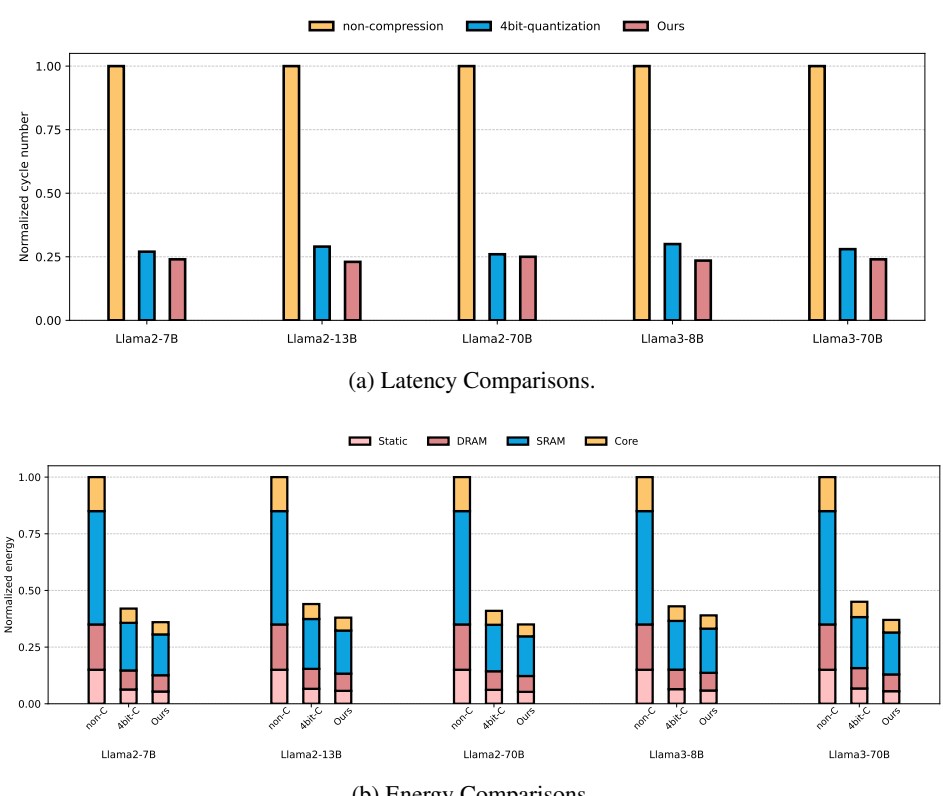

(a) Latency Comparisons.

(b) Energy Comparisons.

Figure 5: Comparisons in different benchmarks.

## CONCLUSION

We present DC-LLM, a novel weight-only compression method that reconstructs each weight block from a compact LFSR seed and a small set of basis tensors, substantially reducing stored weights and chip-to-chip bandwidth. We adapt the number of basis tensors per block with an explained-energy metric to trade off reconstruction error and compression, and we treat block size, seed length, and related hyperparameters as a multi-objective design-space exploration solved via Bayesian optimization. We implement a SystemVerilog RTL accelerator and demonstrate in simulation up to a $4\times$ speedup for memory-bound LLM inference. Extensive experiments on LLM models (7B–70B) show state-of-the-art accuracy at roughly 3–4 bit effective precision. DC-LLM paves the way for new hardware-friendly compression techniques for LLM.

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
