# OpenReview forum: "DC-LLM: HARDWARE-FRIENDLY LLM WEIGHT COMPRESSION VIA DYNAMIC LINEAR COMBINATION"
_ICLR.cc/2026/Conference — ICLR 2026 Conference Withdrawn Submission_

### Official Review · Reviewer_jMYx · 2025-10-31

**Soundness:** 2
**Presentation:** 2
**Contribution:** 2
**Rating:** 2
**Confidence:** 4

**Summary:**

This paper proposes a weight-only compression method for large language models, dynamically constructing each weight block from a seed generated using Linear Feedback Shift Register (LFSR), which is then combined with other generated weight blocks as an approximation of the original model weight. This method increases data-transfer efficiency between memory and compute units by reducing memory footprint. Experiment shows that this method can effectively boost performance on different LLMs of different sizes. Moreover, the paper proposes a dedicated hardware accelerator to speed up the inference process.

**Strengths:**

1. The paper proposes a novel compression method, DC-LLM,  that utilizes LFSR to efficiently generate dense seeds, and develops a calibration-free method to reconstruct weight blocks from these seeds.

2. The paper formulates the process of finding the optimal parameters as a design space exploration (DSE) problem to find parameters that achieve pareto balance.

3.  The paper proposes a seed-compression-aware accelerator design to accelerate inference at hardware level.

**Weaknesses:**

1. Time and memory complexity lacks theoretical analysis. For the reconstruction process, the paper doesn’t provide theoretical analysis that explore the memory and time complexity of the reconstruction process during model inference, making the performance boost less convincing. For the parameter search and weight compression process, the paper also lacks in theoretical analysis, leaving readers to wonder the resource consumption to compress models from scrach.

2. Models used for experiments lack of variety. All experiments conducted in this paper is based on the models of the Llama family, which raises doubts about the effectiveness of the proposed method on other mainstream model families, such as Mistral and Qwen.

3. Experiment setting is vague. The experiment section doesn’t provide readers with the initial settings of the search parameters $T=<B, S, G, R_{th}>$, and the result in table 1 only provides the equivalent quantization level (i.e. 3.8 bit and 2.7 bit). It would be beneficial to provide the initial parameters, as well as the search results of the formulated DSE problem, since it’s now unclear how many set of parameters that reach the pareto balance can be used for different compression rate, which hinders the practicability of the proposed method.

**Questions:**

1. How are the initial hyperparameters $T=<B, S, G, R_{th}>$ defined? Will different initial settings affect the time comsumption of the search process? If so, how do we leverage this problem?
2. Is it possible to implement the hardware design on actual hardwares instead of simulation? If not applicable, what are the obstacles?
3. Can users customize the compression rate easily? In Table 1, the quantization bits of the proposed method  are 3.8 bit and 2.7 bit. Is this value customizable? If so, to what extent of flexibility and granularity?

---

### Official Review · Reviewer_uyW7 · 2025-11-01

**Soundness:** 3
**Presentation:** 3
**Contribution:** 2
**Rating:** 4
**Confidence:** 4

**Summary:**

The paper introduces DC-LLM, a new post-training, weight-only compression method for LLMs. The primary goal is to address the computational and hardware overhead of LLMs , which is mainly caused by the high memory bandwidth required to load model weights.

**Strengths:**

1. Hardware-friendly:
 The proposed algorithm is designed to be efficient on hardware. The choice of a Linear Feedback Shift Register (LFSR) is key, as it's extremely cheap in hardware (just shift and XOR operations)

2. Calibration-Free: The method is "post-training" and does not require any calibration data.

3. Robust Hyperparameter Tuning: The authors treat the compression setup (block size, seed length, etc.) as a formal multi-objective optimization problem. Using Bayesian Optimization to find the best configuration is a standard/robust way to navigate the complex trade-offs between accuracy and storage

**Weaknesses:**

1. Negligible performance gain compared to SeeLM (most important baseline): In performance analysis, authors compared their method with other baselines to show that DC-LLM achieves better performance compared to other baselines, but the performance gain compared to SeedLM which is the most direct related work is negligible.

2. Ablation Study: The paper misses a detailed ablation study, most importantly, authors do not discuss the post-training time complexity and efficiency, given the heavy tuning and calculations, I believe this could be very huge bottleneck/limitation.

3. Custom Hardware is Required for Speedup: most significant barrier to adoption. The paper explicitly states that "conventional GPUs... cannot efficiently reconstruct weights from compact seeds". The 4x speedup is only demonstrated on their custom-designed ASIC accelerator which limits the scalability of the proposed method.

**Questions:**

1. How much improvement can we see in term of Latency compared to other baselines (the ones that you've compared DC-LLM with for performance)?
2. How much time does it take to run the compression?
3. What are the actual numbers for latency if you implement this method and simulate it on CPU instead of GPU?

---

### Official Review · Reviewer_U3q9 · 2025-11-04

**Soundness:** 2
**Presentation:** 1
**Contribution:** 1
**Rating:** 0
**Confidence:** 5

**Summary:**

This paper proposes DC-LLM, a post-training, weight-only compression approach for LLMs. The method reconstructs model weights from compact seeds using an LFSR-based generator that produces pseudo-random basis matrices. Each weight block is expressed as a linear combination of these generated bases, with coefficients chosen adaptively to satisfy an explained-energy threshold. The goal is to reduce storage and memory bandwidth demands while preserving model accuracy. The authors also present a SystemVerilog ASIC design demonstrating up to 4× inference speedup and strong results on LLaMA models compressed to 3–4 bits.
However, the paper’s central idea and much of its formulation are highly similar to SeedLM.

**Strengths:**

- The method addresses a key bottleneck in LLM inference.
- Adaptive basis selection and Bayesian optimization are well-motivated.

**Weaknesses:**

- There is significant conceptual and textual overlap with SeedLM, to the point where many sections appear directly paraphrased from it.
- The novelty claim is weak: DC-LLM’s core idea, representing weight blocks via pseudo-random generator seeds and coefficients, is essentially the same as SeedLM, with only minor algorithmic variations.
- The writing and structure mirror SeedLM closely, suggesting inadequate originality in presentation and contribution framing.
- Empirical results also do not clearly demonstrate substantial improvement over SeedLM to justify a new publication.

**Questions:**

A future version should reflect the authors’ own understanding and narrative framing of the problem, even if the core idea remains related.

**Details Of Ethics Concerns:**

There appears to be substantial textual and structural overlap with the prior work SeedLM. Many sections appear paraphrased or minimally altered, raising potential derivative-submission concerns.

---

### Official Review · Reviewer_oqtn · 2025-11-04

**Soundness:** 3
**Presentation:** 2
**Contribution:** 3
**Rating:** 2
**Confidence:** 3

**Summary:**

The paper presents DC-LLM, a hardware-friendly weight-only compression technique for large language models. The core idea is to use a Linear Feedback Shift Register to generate pseudo-random basis tensors from a seed, and reconstruct each weight block as a linear combination of these bases.
Experiments on LLaMA 2 and 3 (7B–70B) show competitive results with 3–4 bit compression, and the authors further provide a SystemVerilog hardware implementation showing about 4× inference speedup.

**Strengths:**

1.  **Interesting hardware design**: The use of the explained-energy ratio to automatically decide how many bases each weight block needs is elegant. It avoids wasting capacity on easy blocks while keeping difficult ones accurate. The visualization in Fig. 2 nicely illustrates this behavior.
2. **Well-thought system design**:
Framing the compression setup (block size, seed length, threshold, etc.) as a multi-objective problem and solving it via Bayesian optimization is a solid, principled choice. It gives the method a data-driven way to balance accuracy and storage.
3. **Solid Experiments**:
The paper tests on both LLaMA-2 and LLaMA-3 (7B–70B) models with many benchmarks. It’s good to see comparisons against strong baselines (SeedLM, AWQ, OmniQuant, QuIP#) under consistent settings and without fine-tuning or calibration.

**Weaknesses:**

**1. Too hardware-oriented for ICLR:**
The main contributions are focused on implementation efficiency and hardware acceleration. While the work is technically impressive, it feels more suitable for a systems or hardware-focused venue (e.g., MLSys, DAC, ISCA) than for ICLR.

**2. Should add clearer Ablation Study:**
The paper would be stronger with clearer ablations isolating the effects of the adaptive basis selection, the explained-energy ratio threshold, and the Bayesian optimization step. Without them, it’s difficult to assess which components matter most.

**3. Restricted evaluation scope:**
Although the paper includes an extensive set of experiments across multiple LLaMA model sizes (7B–70B) and tasks, all evaluations are confined to the LLaMA family. It remains unclear how well DC-LLM generalizes to other architectures or modalities.

**4. Writing should be polished:**
The paper tries to cover both algorithmic design and hardware implementation, and as a result the main story feels a bit diluted. It’s clear what the authors have achieved, but it takes some effort for the reader to connect all parts together.

**Questions:**

1. Could you add an ablation or some sensitivity analysis on the number of bases or seed length?

2. Do you have any theoretical or empirical insight into why LFSR-generated bases work well for real model weights?

3. How much computation overhead does the reconstruction add in practice, compared to standard quantization?

4. Since the main contribution and validation are heavily hardware-oriented, have you considered submitting or extending this work for a systems or hardware-focused venue (e.g., MLSys, DAC, or ISCA), where it might reach a more appropriate audience and receive stronger technical feedback?

---

> ### Comment · Reviewer_oqtn · 2025-11-13
>
> I find there is a big overlap with SeedLM, so I decide to change my score from 2 to 0.
>
> SEEDLM: COMPRESSING LLM WEIGHTS INTO SEEDS OF PSEUDO-RANDOM GENERATORS is published as a conference paper at ICLR 2025

---

### Note · Authors · 2025-11-13

I have read and agree with the venue's withdrawal policy on behalf of myself and my co-authors.